# Technological Performance and Nutritional Modulation of Bread Enriched with *Cnidoscolus aconitifolius* and *Crotalaria longirostrata* Leaf Flours

**DOI:** 10.3390/plants15010071

**Published:** 2025-12-25

**Authors:** Kimberly Calonico, Esther Pérez-Carrillo, Julian De La Rosa-Millan

**Affiliations:** Centro de Biotecnologia FEMSA, Escuela de Ingenieria y Ciencias, Tecnologico de Monterrey, Av. Eugenio Garza Sada 2501 Sur, Tecnologico, Monterrey 64849, Nuevo Leon, Mexico; perez.carrillo@tec.mx

**Keywords:** functional bread, polyphenol–starch interactions, leaf flours, resistant starch, dietary fiber, enzymatic digestibility

## Abstract

Bread typically exhibits a high glycemic index (GI), motivating interest in plant-based ingredients that can modulate starch digestibility while enhancing nutritional value. This study evaluated the technological, compositional, and digestibility effects of incorporating leaf flours from *Cnidoscolus aconitifolius* and *Crotalaria longirostrata* into wheat bread. Both flours increased protein, dietary fiber, and phenolic content, while modifying dough performance and crumb structure. *C. longirostrata* produced the strongest reduction in predicted glycemic index (pGI), decreasing values by 5.2% on Day 0 and up to 17.8% by Day 5, associated with the highest accumulation of resistant starch. However, this nutritional advantage was accompanied by marked technological drawbacks, including reduced loaf volume and denser crumb. In contrast, *C. aconitifolius* exhibited better technological compatibility, generating breads with higher volume and more cohesive crumb structure, while still achieving meaningful pGI reductions (6.1% on Day 0 and 9.6% by Day 5). Firmness evolution during storage reflected staling-related structural changes but did not involve direct measurement of starch retrogradation. Overall, this work highlights the functional potential of whole leaf flours to enhance the nutritional profile and glycemic behavior of bread, while underscoring the formulation-dependent trade-offs that influence technological quality. These findings provide a foundation for developing optimized, lower-glycemic baked products using underutilized botanical ingredients.

## 1. Introduction

Bread is a staple food consumed worldwide, yet its typically high glycemic index (GI) has raised concerns regarding its contribution to metabolic disorders such as insulin resistance, type 2 diabetes, and obesity [1,2]. Standard white wheat breads generally exhibit GI values of 70 to 100, depending on formulation, structural characteristics, and processing conditions [3,4]. These values classify bread as a rapidly digestible carbohydrate source that can elicit pronounced postprandial glycemic responses, underscoring the need for strategies to modulate starch digestibility and reduce its glycemic impact. The GI of bread is determined by multiple factors, including starch composition and structure, protein-starch interactions, dietary fiber content, and the rate of enzymatic hydrolysis during digestion [5,6].

In recent years, there has been increasing interest in reformulating bread with plant-based ingredients rich in dietary fiber and polyphenols to enhance nutritional value and modulate starch digestibility. Widely studied examples include Moringa oleifera, kale (*Brassica oleracea* var. *acephala*), spinach (*Spinacia oleracea*), and amaranth leaves (*Amaranthus* spp.), whose incorporation into baked products has been shown to improve antioxidant capacity, increase resistant starch, and reduce glycemic response [7,8,9]. This growing body of evidence highlights the broader relevance of exploring botanical ingredients as functional components within cereal-based foods.

Among emerging plant-based ingredients, *Cnidoscolus aconitifolius* (chaya) and *Crotalaria longirostrata* (chipilín) are traditional Mexican plants recognized for their high levels of dietary fiber, protein, minerals, and phenolic compounds, rather than for medicinal use in the clinical sense [10,11]. While these plants have been historically consumed as vegetables, recent studies using leaf extracts have reported antioxidant, anti-inflammatory, and enzyme-inhibitory activities, particularly against α-amylase and α-glucosidase, enzymes central to starch digestion [12,13,14,15,16]. However, leaf flours differ substantially from extracts in composition, compound concentrations, and functional behavior within dough. Therefore, conclusions drawn from extract-based studies cannot be directly extrapolated to food applications. A detailed characterization of whole-leaf flours is needed to evaluate their technological and nutritional impact in bakery systems.

An essential distinction in this work is the use of whole leaf flours rather than extracts, as flours retain dietary fiber, cell wall structures, and bound phenolics that interact with gluten development, water distribution, and starch-enzyme accessibility in ways that extracts cannot replicate. This matrix-driven behavior is relevant because several established strategies to reduce glycemic response in bread, such as sourdough fermentation, resistant starch enrichment, enzymatic modification, or incorporation of fruit and vegetable powder, rely on altering starch structure or inhibiting digestive enzymes. Positioning leaf flours within this broader landscape highlights both their potential as functional ingredients and the practical challenges associated with their use, including effects on dough handling and loaf volume, as well as potential gastrointestinal tolerance issues for some consumers.

Dietary fiber and polyphenols play well-documented roles in modulating starch digestibility. Soluble fiber increases viscosity in the gastrointestinal tract and slows enzymatic hydrolysis, whereas insoluble fiber can physically limit enzyme accessibility to starch granules [2,17]. Polyphenols may further reduce starch digestibility by binding to digestive enzymes or forming starch-polyphenol complexes that hinder gelatinization and hydrolysis [13,18]. These compositional effects can promote starch retrogradation and increase resistant starch formation, features associated with lower glycemic response, while simultaneously influencing loaf volume, crumb texture, and overall bread quality [9,19,20].

Although both *C. aconitifolius* and *C. longirostrata* have been studied for their biological activities, no research has evaluated the incorporation of their leaf flours into wheat bread nor examined their combined effects on dough rheology, technological performance, starch digestibility, and predicted glycemic index. Likewise, no information is available on how the structural and compositional properties of these flours influence fermentation dynamics and bread staling.

The present study aims to characterize the technological, structural, and nutritional effects of incorporating leaf flours from *C. aconitifolius* and *C. longirostrata* into wheat bread. Specifically, we evaluate their impact on dough properties, bread volume and texture, crumb structure, color, dietary fiber content, phenolic content, starch digestibility fractions, and predicted glycemic index (pGI). By analyzing whole flours rather than extracts, this work provides mechanistic insights into how botanical matrices influence bread quality and glycemic potential, contributing to broader efforts to design functional, lower-GI cereal products while recognizing the technological trade-offs inherent to these ingredients.

## 2. Results

### 2.1. Chemical Composition of Breads

The proximate composition of the bread samples is presented in Table 1. Protein content increased significantly in both plant-enriched formulations compared with the control (10.13 ± 0.27%), reaching 12.24 ± 0.10% in *C. aconitifolius* bread and 12.53 ± 0.10% in *C. longirostrata* bread. The incorporation of leaf flours also resulted in a dilution of total starch, which decreased from 65.6 ± 0.30% in the control bread to approximately 62% in both substituted formulations.

Both enriched breads exhibited higher dietary fiber levels than the control. Soluble fiber increased from 1.26 ± 0.15% in the control to 2.22 ± 0.10% in *C. aconitifolius* and 2.26 ± 0.08% in *C. longirostrata* breads. Insoluble fiber followed a similar trend, increasing from 3.15 ± 0.13% in the control to 4.11 ± 0.09% and 4.09 ± 0.10%, respectively, reflecting the naturally higher fiber content of the plant flours.

Lipid content rose moderately with the addition of the leaf flours, from 5.33 ± 0.11% in the control to 6.08 ± 0.49% in *C. aconitifolius* and 6.32 ± 0.10% in *C. longirostrata* breads. A similar increase was observed in ash content, which rose from 5.44 ± 0.27% in the control to 6.24 ± 0.12% and 6.36 ± 0.17%, respectively. These increases are consistent with the higher mineral and lipid contributions of the plant materials [14].

### 2.2. Molecular Changes During Fermentation Dynamics

The incorporation of the leaf flours markedly increased the total phenolic content of the breads (Table 2). The control bread contained only trace amounts of phenolic compounds (0.01 ± 0.00 mg GAE/100 g). In contrast, breads enriched with *C. aconitifolius* and *C. longirostrata* reached 9.56 ± 1.21 mg GAE/100 g and 19.82 ± 0.90 mg GAE/100 g, respectively. The substantially higher phenolic content observed in *C. longirostrata* bread reflects the larger phenolic load previously reported for this leaf flour.

The addition of the plant flours also modified the levels of free sugars and Free Amino Nitrogen (FAN). Free sugar content was highest in the control bread (11.5 ± 0.38%) and decreased in the enriched formulations, registering 10.53 ± 0.36% in *C. aconitifolius* and 9.54 ± 0.13% in *C. longirostrata* breads. This reduction may result from interactions between polyphenols and fiber with starch and enzymes, potentially slowing starch hydrolysis during fermentation.

In contrast, FAN values increased in both enriched breads. Levels rose from 62.41 ± 1.96 mg/100 g in the control to 68.31 ± 1.55 mg/100 g in *C. aconitifolius* and 74.30 ± 2.18 mg/100 g in *C. longirostrata* breads, suggesting enhanced proteolysis during fermentation or additional nitrogen-containing compounds contributed by the leaf flours. These changes in FAN and sugar availability may influence yeast activity and fermentation dynamics.

### 2.3. Technological Properties of Bread

The physical and structural characteristics of the breads are presented in Table 3 and Figure 1. The control bread showed the greatest loaf volume (701.25 ± 42.69 cm^3^), consistent with optimal gluten development and gas retention. In contrast, bread containing *C. longirostrata* exhibited the lowest post-baking height (7.15 ± 0.19 cm) and volume (521.25 ± 11.08 cm^3^), which corresponded to the highest apparent density value (0.29 ± 0.00 g/cm^3^). These reductions indicate that the incorporation of *C. longirostrata* flour markedly impaired dough expansion during proofing and baking.

Differences in final loaf weight were also observed among samples (Table 3). The control bread exhibited a slightly lower final weight, consistent with greater moisture loss during baking. This behavior is expected for breads with higher loaf volume and a more open crumb structure, as greater gas expansion facilitates water evaporation during heating [21,22]. In contrast, breads enriched with *C. aconitifolius* and *C. longirostrata* retained more moisture, likely due to the higher water-binding capacity and swelling ability of their dietary fiber fractions. Similar effects have been reported in breads formulated with plant-based fibers or phenolic-rich ingredients, which enhance water retention and reduce baking losses by strengthening the crumb matrix and decreasing vapor diffusivity [9,23].

The control bread also showed the highest proportion of alveoli (40.31 ± 1.33%). The addition of *C. aconitifolius* significantly reduced alveolation to 29.53 ± 0.64%, suggesting partial disruption of the gluten network. *C. longirostrata*, despite producing lower loaf volume, displayed 38.28 ± 2.02% of alveoli. Image analysis (Figure 1B) revealed that this apparent alveolar quantity did not translate into an open crumb structure; instead, *C. longirostrata* bread showed compact, irregular regions with partially collapsed air cells. These structural defects are consistent with dough weakening caused by the higher fiber content and phenolic composition of this flour.

### 2.4. Color

The color characteristics of the breads are presented in Table 4 and Figure 2. The control bread exhibited the highest lightness value (L* = 78.42 ± 3.67), reflecting the typical appearance of standard wheat bread crumb. In contrast, the incorporation of leaf flours resulted in darker crumbs due to the natural pigments and phenolic compounds present in the plant materials. Bread containing *C. aconitifolius* showed the greatest shift toward redness (a* = 7.55 ± 0.52), consistent with the pigmentation associated with this leaf flour. *C. longirostrata* produced the highest yellowness value (b* = 28.12 ± 1.23), indicative of its characteristic carotenoid and chlorophyll-derived color components.

Among the enriched breads, *C. longirostrata* exhibited the largest total color difference relative to the control (ΔE = 420.66), confirming that its incorporation results in the most pronounced visual modification of the crumb. These color changes reflect both the intrinsic pigments of the plant flours and their interactions with Maillard reaction products formed during baking.

### 2.5. Firmness and Staling Behavior

The evolution of crumb firmness during storage is presented in Figure 3. On Day 0, bread containing *C. longirostrata* exhibited the highest initial firmness, indicating a denser and less extensible crumb structure immediately after baking. In contrast, *C. aconitifolius* bread showed the lowest firmness at this stage, suggesting better gas retention or a more elastic crumb matrix despite its reduced loaf volume.

By Day 3, a pronounced increase in firmness was observed in *C. aconitifolius* bread, causing it to surpass *C. longirostrata*. This behavior suggests accelerated staling, likely associated with faster moisture redistribution and structural reorganization of starch and gluten. Although increases in firmness are commonly interpreted as manifestations of starch retrogradation, it is important to note that retrogradation itself was not directly measured in this study; instead, firmness evolution serves as an indirect indicator of staling-related processes occurring within the crumb matrix.

By Day 5, *C. aconitifolius* maintained higher firmness than the control bread, whereas *C. longirostrata* remained the softest among all samples and showed the smallest firmness increase between Days 3 and 5. The slower firming rate in *C. longirostrata* bread may reflect its disrupted gluten structure and higher water-binding capacity, which together can delay crumb hardening during storage.

Although bread stored for five days is not suitable for sensory consumption unless preserved or toasted, the purpose of evaluating samples up to Day 5 was strictly technological. Storage intervals of 3–5 days are widely used in staling and shelf-life research to characterize textural degradation and to understand its relationship with starch-protein interactions and digestibility dynamics [10].

### 2.6. In Vitro Digestible Starch and Predicted Starch Glycemic Index (pGI)

The distribution of starch digestibility fractions and the corresponding predicted glycemic index (pGI) values are presented in Figure 4 and Table A1. On Day 0, the control bread showed the highest pGI (92.21 ± 0.25), consistent with its high proportion of rapidly digestible starch (RDS, 88.65 ± 0.35%). In contrast, both plant-enriched breads exhibited lower RDS values and higher levels of resistant starch (RS), indicating slower initial starch hydrolysis. During storage, RS increased markedly in both enriched formulations, reflecting progressive starch retrogradation. By Day 5, *C. longirostrata* bread exhibited the highest RS content (54.85 ± 8.19%), which corresponded to the lowest pGI (74.41 ± 2.61). This substantial shift suggests that the high fiber and phenolic content of *C. longirostrata* flour enhanced the formation of retrograded starch fractions. *C. aconitifolius* bread also showed an increase in RS over storage, reaching 31.56 ± 2.93% on Day 5 and reducing its pGI to 81.91 ± 0.97. Although less pronounced than *C. longirostrata*, this effect still indicates a beneficial moderation of starch digestibility.

### 2.7. Multivariate Overview of Bread Attributes

To visualize overall patterns among formulations, hierarchical clustering and PCA were used purely descriptively due to the limited number of samples. The clustering dendrograms (Figure 5A,C,E) highlight broad similarities and differences without implying statistical separation. On Day 0, the control bread was grouped independently from the plant-enriched formulations, reflecting its higher starch digestibility and lower fiber–phenolic content. As storage progressed, the sample containing *C. longirostrata* gradually shifted away from both the control and *C. aconitifolius* breads, forming a separate branch by Day 5. This qualitative divergence is consistent with the distinct moisture retention, crumb-softening behavior, and starch-protein interactions observed earlier, rather than with direct measurements of fermentation or retrogradation.

The PCA loading plots (Figure 5B,D,F) further summarize the relationships among variables across storage. By Day 5, resistant starch (RS) accounted for much of the variation along the first principal component (PC1), aligning with increased firmness, lower predicted GI, and reduced free sugar content. These associations indicate that storage-driven structural reorganization of starch, inferred from texture and digestibility changes, was the dominant factor shaping sample distribution in the multivariate space. Phenolic content contributed minimally to sample separation, consistent with its relatively stable behavior during storage compared with the more dynamic evolution of starch digestibility and texture. Overall, the multivariate analyses serve only to consolidate trends already described in previous sections and are not intended to provide additional statistical inference.

## 3. Discussion

### 3.1. Chemical Composition of Breads

The increases in protein content observed in the enriched breads compared with the control can be attributed to the naturally higher protein levels present in both *C. aconitifolius* and *C. longirostrata* flours. This elevation not only enhances the nutritional value of the breads but may also contribute to dough development by increasing the availability of nitrogen-containing compounds involved in fermentation and gluten interactions [24,25]. The higher dietary fiber content in both formulations, encompassing soluble and insoluble fractions, is consistent with the known composition of these leafy flours. Dietary fiber is recognized for its role in promoting satiety, supporting gastrointestinal function, and moderating postprandial glycemic response [17,26] findings that align with the digestibility results obtained in this study.

The moderate increases in fat and ash contents in the enriched breads reflect the intrinsic lipid and mineral composition of the plant flours. These increases suggest that incorporating such ingredients may also contribute beneficial micronutrients and support improved mineral intake [7]. Similar compositional enhancements have been reported in studies that incorporate polyphenol- and fiber-rich plant materials into cereal products, where these ingredients not only improve nutrient density but also influence dough behavior and bread structure [26].

These compositional changes corresponded with modifications in the technological properties of the breads. For example, the denser crumb and higher firmness observed in *C. aconitifolius* bread may be linked to interactions between plant-derived fibers, phenolic compounds, and the gluten-starch matrix. Such interactions can strengthen certain structural regions while limiting gas expansion, ultimately affecting texture. These effects highlight the need to carefully optimize substitution levels to balance the functional and nutritional benefits of plant flours with their impact on rheological performance, crumb structure, and overall sensory acceptability.

Although the contrasting behaviors of *C. aconitifolius* and *C. longirostrata* naturally raise the question of whether a combined formulation (e.g., 2.5% + 2.5%) could balance their respective advantages, the present study was intentionally designed to isolate the individual contributions of each ingredient. Blending the flours would have simultaneously altered phenolic composition, fiber type, and water-binding capacity, making it difficult to attribute specific technological or nutritional outcomes to a particular component. Identifying these ingredient-specific mechanisms is a necessary first step before evaluating potential synergistic or compensatory interactions in mixed systems. Future studies will therefore assess blended formulations to determine whether partial substitution of each flour can optimize loaf structure, moisture retention, resistant starch formation, and glycemic response.

### 3.2. Molecular Changes During Fermentation Dynamics

The substantial increase in total phenolic content (TPC), particularly in the *C. longirostrata* formulation (19.82 ± 0.90 mg GAE/100 g), reflects the inherently high polyphenol concentration of this leaf flour. These elevated phenolic levels may enhance the bread’s antioxidant potential and contribute to physiological effects, such as attenuating oxidative stress and moderating glycemic response [8,27]. The higher TPC in *C. longirostrata* bread is likely driven by its larger abundance of tannins, flavonoids, and other water-extractable phenolics. These compounds are known to form complexes with gluten proteins and starch granules during mixing and baking, potentially modifying dough rheology and reducing starch accessibility to digestive enzymes [5,28].

The reduction in free sugar content observed in the enriched breads, with *C. longirostrata* reaching the lowest value (9.54 ± 0.13%), may be partly due to the inhibitory effects of phenolics on amylolytic enzymes during fermentation. By limiting the hydrolysis of starch into fermentable sugars, polyphenols can reduce yeast’s access to fermentable sugars and, subsequently, the availability of sugars for Maillard reactions during baking [7,29]. This reduction in free sugars is consistent with the lower RDS levels and reduced predicted glycemic index observed in both enriched formulations.

In contrast, the increased Free Amino Nitrogen (FAN) values in the enriched breads (up to 74.3 ± 2.18 mg/100 g) indicate enhanced proteolysis or additional nitrogenous compounds contributed directly by the leaf flours. Higher FAN levels provide readily assimilable nitrogen for yeast metabolism, which may affect fermentation dynamics, gas production, and dough expansion [18,22]. The presence of small peptides and free amino acids in the plant flours likely contributes to this increase. Taken together, the changes in TPC, free sugars, and FAN demonstrate that the biochemical composition of the leaf flours influences both fermentation processes and the nutritional and functional characteristics of the resulting breads.

### 3.3. Technological Properties of Bread

The technological results reveal a clear structural trade-off associated with the incorporation of the leaf flours. The lowest loaf volume (521.25 ± 11.08 cm^3^) and highest density (0.29 ± 0.00 g/cm^3^) observed in *C. longirostrata* bread are consistent with substantial disruption of the gluten network caused by its high insoluble fiber content. Insoluble fiber particles can interrupt the continuity of the gluten-starch matrix, reducing its ability to retain gas during proofing and baking. This structural weakening is corroborated by cross-sectional images showing a compact, irregular crumb with partially collapsed air cells, characteristic of dough systems with insufficient viscoelastic strength [30,31,32].

In contrast, *C. aconitifolius* exhibited better technological compatibility. Its incorporation resulted in a significantly higher loaf volume (597.5 ± 77.08 cm^3^) than that achieved with *C. longirostrata*, suggesting a less severe interference with gluten development. Although *C. aconitifolius* reduced alveoli to 29.53 ± 0.64%, this outcome may reflect its higher soluble fiber fraction, which enhances water retention and can provide moderate stabilization of the gluten network during baking. Soluble fibers are known to increase dough viscosity and reduce gas cell coalescence, thereby partially offsetting their diluting effect on gluten. Overall, the results indicate that *C. aconitifolius* flour integrates more successfully into breadmaking systems, whereas *C. longirostrata* exerts a substantially greater weakening effect on dough structure.

### 3.4. Color

The reduction in lightness (L*) observed in both enriched breads is an expected consequence of incorporating pigmented flours rich in polyphenols. Polyphenolic compounds readily undergo oxidation and participate in Maillard-type reactions during baking, contributing to darker crumb coloration [27,33]. The distinct chromatic shifts recorded in the a* and b* coordinates further reflect the unique pigment profiles of the two plant flours. The higher redness (a*) observed in *C. aconitifolius* bread and the increased yellowness (b*) in *C. longirostrata* bread correspond to their respective compositions in chlorophyll derivatives, flavonoids, and carotenoid pigments [34,35].

The substantial total color difference (ΔE) measured for *C. longirostrata* indicates that this flour exerts a particularly strong influence on bread coloration. This effect is consistent with its higher phenolic concentration and pigment diversity, which together intensify the deviation from the control crumb appearance. Similar findings have been reported in breads and cereal products enriched with other polyphenol-rich ingredients, where the addition of plant-derived pigments leads to pronounced alterations in color attributes [32,36].

### 3.5. Texture Characteristics

The textural evolution of the breads enriched with *C. aconitifolius* and *C. longirostrata* reflects the influence of their distinct compositional and structural characteristics. The pronounced increase in firmness observed in *C. aconitifolius* bread during storage suggests the formation of a denser crumb structure, likely driven by reduced gas retention and a stronger, yet less extensible, gluten-fiber matrix. This trend can be explained by starch retrogradation and moisture redistribution within the crumb, two well-established mechanisms governing bread staling [27,36].

In contrast, *C. longirostrata* bread maintained consistently lower firmness throughout storage, indicating a slower staling rate. This behavior is consistent with its highly disrupted gluten network, which limits initial structural development but may also reduce the extent of starch retrogradation. Additionally, its formulation appears to retain moisture more effectively, helping to delay crumb hardening over time [29,36].

These observations align with previous reports showing that flours rich in fiber and polyphenols can modify staling kinetics by altering water mobility, gluten interactions, and starch recrystallization patterns [35]. Overall, compositional differences among plant flours play a decisive role in determining the firmness and aging behavior of the breads.

### 3.6. In Vitro Digestible Starch and Predicted Starch Glycemic Index (pGI)

The incorporation of *C. aconitifolius* and *C. longirostrata* into bread formulations significantly modified starch digestibility patterns and reduced the predicted glycemic index (pGI). On Day 0, the control bread presented the highest pGI, consistent with its elevated proportion of rapidly digestible starch (RDS), which is hydrolyzed quickly and contributes to higher postprandial glucose responses [37,38]. In contrast, both enriched breads displayed lower pGI values at this stage, driven by increased proportions of slowly digestible starch (SDS) and resistant starch (RS). These fractions exert a moderating effect on starch hydrolysis and are associated with a more gradual rate of glucose release during digestion [38].

Storage further amplified these effects. Both plant-enriched breads showed notable increases in RS content over time, with *C. longirostrata* exhibiting the most significant increase. This increase reflects enhanced starch retrogradation during cooling and storage, thereby reducing enzymatic susceptibility and lowering pGI. The strong retrogradation effect observed in *C. longirostrata* may result from the combined action of its higher phenolic content and fiber composition. Polyphenols can form complexes with starch granules, limiting enzyme access, while soluble fiber enhances water retention and promotes the formation of retrograded amylose and amylopectin structures during storage [23,36].

From a biological standpoint, the relevance of the pGI reductions observed in this study must be interpreted beyond statistical significance. Even a modest decrease in pGI, from 92 to 87, as seen with *C. aconitifolius* can translate into a measurable attenuation of postprandial glucose excursions, particularly for individuals with insulin resistance or impaired glucose tolerance. According to established GI classifications, foods within the high-GI category (>70) may still produce lower glycemic responses when reduced by 5–10 units, which has been associated with improved metabolic outcomes in controlled feeding trials [31,32]. Although pGI values derived from in vitro digestion do not directly quantify human glycemic responses, they reliably capture relative differences among formulations and indicate physiologically meaningful shifts in starch digestibility. In this context, the larger reduction produced by *C. longirostrata* (up to ~18% by Day 5) suggests potentially greater metabolic relevance, consistent with its higher resistant starch content and slower digestion profile.

### 3.7. Multivariate Visualization of Technological and Nutritional Attributes

To provide a complementary overview of sample behavior, PCA and hierarchical clustering were used qualitatively to summarize relationships among the technological and nutritional variables. Given the limited number of formulations, these analyses are not intended for statistical inference, but rather to consolidate trends described earlier in the Results.

Across storage days, the control bread consistently separated from the plant-enriched samples, reflecting its higher starch digestibility and lower fiber and phenolic content. By contrast, *C. longirostrata* gradually separated from both the control and *C. aconitifolius* formulations, consistent with its distinct moisture retention, slower firming pattern, and greater accumulation of resistant starch. These qualitative groupings reinforce the main findings on technological performance and starch digestibility but do not provide additional mechanistic insights.

## 4. Materials and Methods

### 4.1. General

To ensure the reproducibility of the study, all materials, data, and protocols are available upon request. No human or animal subjects were used in this study; thus, no ethical approval was required (as the pGI estimation was conducted in vitro). Generative Artificial Intelligence (GenAI) was not used for study design, data collection, analysis, or interpretation, but was employed for minor text editing (grammar and syntax optimization).

### 4.2. Preparation and Characterization of Plant Leaf Flours

Fresh leaves of *C. aconitifolius* and *C. longirostrata* were obtained from a supermarket in Monterrey, Nuevo León, Mexico (Supermercados H-E-B). Mature, undamaged leaves were selected to ensure compositional consistency. Leaves were washed with distilled water, drained, and dried in a forced-air convection oven (Binder GL-45, Dresden, Germany) at 55 °C for 12 h to minimize degradation of phenolic compounds while producing stable leaf flours. Dried leaves were then milled using a Wiley knife mill (Thomas Scientific, Swedesboro, NJ, USA) and passed through a 500 µm mesh. The resulting material is referred to as leaf flour, in line with common terminology used in cereal and bakery applications.

The proximate composition, dietary fiber profile, and phenolic content of these flours have been previously characterized in detail by Calonico et al. [5], and the analytical procedures followed here were used to maintain methodological consistency. Comparable approaches for processing and characterizing leafy vegetable powders intended for food applications have been described by Jiménez-Aguilar & Grusak [7] and Wang et al. [10], further supporting the reliability of the preparation procedures used in this work. The preparation steps applied in the present study, washing, controlled thermal drying, and standardized milling, also align with established protocols for producing functional plant flours as reported by Chen et al. [11] and Yaver & Bilgiçli [9], which emphasize the importance of controlled drying and particle size uniformity for preserving phenolic integrity and ensuring reproducible functional behavior.

All flour batches were prepared under identical conditions and stored at 4 °C in opaque, airtight polyethylene containers for no more than two weeks before use, following best practices described in the studies above. This ensured minimal moisture uptake, oxidative degradation, or compositional variability before further analysis and bread formulation.

### 4.3. Bread Formulation and Baking Procedure

The use of a 5% substitution level was selected based on both prior literature and preliminary formulation trials, as well as the intrinsic composition of the two leaf flours used. Leaf flours such as *C. aconitifolius* and *C. longirostrata* contain high levels of dietary fiber, insoluble structural material, and phenolic compounds, which rapidly weaken the gluten network even at low inclusion percentages. Previous studies incorporating polyphenol and fiber-rich plant materials into bread commonly report that levels above 5–7% result in substantial reductions in loaf volume, dense crumb structure, and impaired gas retention. Therefore, 5% is widely used as an upper limit for testing structurally disruptive plant ingredients in yeast-leavened products.

A straight-dough method, adapted from AACC baking procedures, was used to prepare the bread loaves [39]. The formulation is presented in Table 5. All doughs were prepared under identical conditions. Dry ingredients were first blended for 1 min, after which water and yeast were added. Mixing was performed in a planetary mixer (KitchenAid, Benton Harbor, MI, USA) for 2 min at low speed, followed by 6 min at medium speed to ensure gluten development. The resulting doughs were fermented at 30 °C and 85% relative humidity for 60 min. After bulk fermentation, the doughs were manually degassed, molded into cylindrical pieces of 350 g, and placed in standard metal pans (19 × 10 × 9 cm).

A final proofing step was conducted at 30 °C and 85% relative humidity for 45 min. Loaves were baked in a convection oven (Whirlpool, Benton Harbor, MI, USA) at 180 °C for 25 min or until an internal temperature of 96 to 98 °C was reached, measured with a calibrated thermocouple. After baking, the loaves were removed from the pans and cooled at room temperature (22 ± 1 °C) for 30 min.

### 4.4. Storage Conditions of Bread Samples

After cooling, each loaf was packaged in a low-density polyethylene bag (50 µm thick) to limit moisture loss and prevent surface dehydration. All samples were stored at 22 ± 1 °C under controlled ambient conditions and protected from direct light exposure by placing them inside opaque plastic containers. No refrigeration was used at any point to avoid altering the kinetics of starch retrogradation. Storage time points for analysis were Day 0 (freshly baked), Day 3, and Day 5. For each time point, separate loaves were used to avoid repeated handling or exposure that could affect moisture content or texture.

### 4.5. Loaf Volume, Height, and Apparent Density

Loaf volume, height, and apparent density were measured to evaluate the macroscopic structural properties of the breads. After cooling for 30 min at room temperature, each loaf was removed from the pan, and its center height was measured with a digital caliper (±0.01 mm accuracy). Loaf volume was determined using the rapeseed displacement method, as described in AACC Method 10-05.01 [40]. Briefly, the loaf was placed in a calibrated-volume chamber, and the displaced rapeseed volume was recorded as the loaf volume. Loaf density (g/cm^3^) was calculated by dividing the loaf weight by its measured volume. All measurements were performed in triplicate using independent loaves.

### 4.6. Color Measurement (CIE Lab)*

Color measurements were performed on 15 mm thick slices taken from the central region of each loaf to avoid crust interference. A Nix Color Sensor (Nix Sensors Ltd., Hamilton, ON, Canada), a portable spectrophotometric device calibrated prior to each session using the manufacturer’s white reference tile, was used for all measurements. Slices were placed on a standardized matte-white background under uniform illumination from a 6500 K LED light source, ensuring consistent diffuse lighting and minimizing shadows. Color coordinates were recorded in the CIE Lab* system, where L* represents lightness, a* the red-green axis, and b* the yellow-blue axis. Five independent readings were taken per sample to ensure measurement repeatability.

The total color difference (ΔE) between each enriched bread and the control bread was calculated using Equation (1).(1)ΔE=(∆L2+∆a2+(∆b)2)12

### 4.7. Crumb Microstructure Analysis (ImageJ)

Image analysis was conducted using ImageJ software (version 1.53, National Institute of Health, Bethesda, MD, USA), an open-source image processing program widely used for quantitative analysis of microstructural features in food products. The software supports threshold-based segmentation, binary image generation, particle detection, and morphological parameter measurement, making it suitable for evaluating crumb porosity and cell structure. Analyses were performed on 15 mm thick crumb slices. Images were captured with a high-resolution camera (iPhone 14, 12 MP, Apple Inc., Cupertino, CA, USA) mounted on a fixed tripod at 25 cm. Illumination was standardized using a 6500 K LED light source positioned at 45 degrees, and samples were placed on a matte white background to minimize reflections. A millimeter scale was included in each image to enable spatial calibration.

Images were imported into ImageJ and converted to 8-bit grayscale. A uniform threshold was applied to segment gas cells from the surrounding crumb matrix, and binary masks were generated. The “Analyze Particles” function was then used to quantify cell number, average cell diameter, and size distribution. For consistency across all samples, the region of interest was restricted to a 400 × 400 pixel area centered on each slice. All measurements were performed in triplicate using independent slices from separate loaves.

### 4.8. Texture Analysis and Firmness Evaluation

Bread crumb firmness, used as an indicator of staling, was measured using a TVT 6700 Texture Analyzer (Perten Instruments, Hägersten, Sweden) operated with TextCalc software version 5.2.0.243. Slices measuring 5 × 5 cm and 15 mm in thickness were cut from the center of each loaf to avoid crust interference and ensure sample uniformity. Measurements were performed using a cylindrical stainless-steel probe (40 mm diameter, 65 mm height). The probe compressed each sample to 40% of its original height at a constant speed of 1.7 mm/s, applying a single compression cycle in accordance with standard procedures for bread crumb evaluation. Firmness values were recorded at three storage times: Day 0 (freshly baked), Day 3, and Day 5. Independent loaves were used for each storage day to prevent repeated deformation of the samples. Firmness was selected as the primary indicator of staling because it reflects the progressive crumb hardening associated with starch retrogradation and moisture redistribution during storage, processes widely recognized as the main physicochemical mechanisms responsible for bread aging.

### 4.9. Chemical Composition Analysis

Bread samples were prepared for chemical analysis by grinding the crumb portion using a Wiley knife mill (Thomas Scientific, Swedesboro, NJ, USA) fitted with a 1 mm^2^ screen. The milled material was thoroughly homogenized and stored in airtight polyethylene containers at room temperature until analysis. All compositional data are expressed on a dry basis; samples were analyzed in their natural state and converted to dry basis through moisture correction during calculation, rather than by pre-drying, to avoid altering nutrient or phenolic composition.

Moisture and ash contents were determined following AACC Method 44-15.02 and AACC Method 08-17.01 [40], respectively. Crude protein content (N × 6.25) and Lipid content (ether extract) was measured according to AOAC [16] and AACC Method 32-10.01 [40], respectively. Total dietary fiber, insoluble dietary fiber, and soluble dietary fiber were analyzed using the Megazyme K-TDFR kit (Megazyme, Wicklow, Ireland), following the procedures detailed in AOAC Method 991.43 and AACC Method 32-07.01 [40]. All measurements were performed in triplicate using independently milled samples to ensure analytical precision and reproducibility.

### 4.10. Phenolic Content, Free Sugars, and Free Amino Nitrogen

Total phenolic compounds (TPC) were quantified using the Folin–Ciocalteu colorimetric assay. Phenolic extraction was performed following a hot aqueous extraction protocol adapted from Cabello-Pasini et al. [22] with minor modifications. For each sample, 5 g of milled bread crumb (control, *C. aconitifolius*, or *C. longirostrata*) was mixed with 50 mL of distilled water in a glass beaker. The mixture was heated at 90 °C for 5 min to promote the release of water-soluble phenolic compounds without causing excessive thermal degradation. After heating, the suspension was filtered through Whatman No. 1 filter paper, and the filtrate was centrifuged at 1000× *g* for 15 min to remove residual solids. The resulting supernatant was collected and stored in amber bottles at 4 °C until analysis. For TPC determination, aliquots of the extract were mixed with Folin–Ciocalteu reagent and sodium carbonate solution following standard assay conditions. After incubation at room temperature for color development, absorbance was measured at 620 nm using a UV-Vis spectrophotometer. Phenolic concentration was calculated from a gallic acid calibration curve and expressed as mg gallic acid equivalents (GAE) per 100 g of sample [22].

### 4.11. Soluble Sugars Profile

Soluble sugars were quantified using the method of Karkacier et al. [37] with minor modifications. Approximately 50 mg of milled bread crumb was frozen with liquid nitrogen and ground to a fine powder to prevent enzymatic activity during extraction. The powder was homogenized in 1 mL of nanopure water and vortexed thoroughly. The homogenate was centrifuged at 10,000 rpm for 15 min, and the resulting supernatant was filtered through a 0.45 µm membrane to remove particulates before chromatographic analysis. Filtrates were analyzed by HPLC using a Waters system equipped with a MetaCarb 87C carbohydrate analysis column and an appropriate guard column. Deionized water served as the mobile phase at a flow rate of 0.5 mL/min. Detection was performed with a Waters 410 differential refractometer, and quantification was carried out using Empower 3.7 software. Calibration curves were generated using analytical-grade standards of fructose, glucose, and sucrose.

### 4.12. Fractions of Digestion of Starch

In vitro starch digestibility fractions were determined according to the method of Englyst et al. [38], with minor adaptations. Milled bread samples (400 mg) were rehydrated in 10 mL of deionized water and heated to boiling for 20 min with intermittent vortex mixing to ensure complete starch gelatinization. After heating, the mixtures were cooled to 37 °C, and gastric digestion was initiated by adding 8 mL of pepsin solution (5.21 mg/mL). Samples were incubated in a shaking water bath at 37 °C for 2 h.

Subsequently, the pH was adjusted to 5.2 using sodium acetate buffer (0.5 M), and intestinal digestion was performed by adding a mixed enzyme solution containing pancreatin, amyloglucosidase, and invertase. Aliquots (1 mL) were collected at 20 and 120 min to quantify glucose released during hydrolysis. Glucose concentrations were determined using a glucose oxidase-peroxidase colorimetric assay.

Starch digestibility fractions were calculated as follows:RDS (%) = (G20 − FG) × 0.9 × 100/TSSDS (%) = (G120 − G20) × 0.9 × 100/TSRS (%) = 100 − RDS − SDS
where G20 and G120 represent glucose concentrations at 20 and 120 min, respectively; FG is free glucose in the sample before digestion; and TS corresponds to total starch content.

### 4.13. Estimated Starch Glycemic Index (pGI)

The predicted glycemic index (pGI) of the bread samples was determined using the in vitro starch hydrolysis protocol described by Granfeldt et al. [41]. This method simulates the early stages of gastrointestinal digestion by monitoring the rate and extent of starch hydrolysis by porcine pancreatic α-amylase. Milled bread samples were incubated with the enzyme mixture, and aliquots were collected at 30, 60, 90, 120, and 180 min. The amount of hydrolyzed starch at each time point was quantified using a glucose oxidase-peroxidase assay kit. A hydrolysis curve was constructed for each bread type by plotting the % of hydrolyzed starch against time. The hydrolysis index (HI) was calculated as the ratio between the area under the hydrolysis curve of the test sample and that of a reference bread (commercial white bread), integrated over the same time interval. The predicted glycemic index was then estimated using the Goñi regression equation:pGI = 39.71 + 0.549 × HI
where HI represents the hydrolysis index expressed as a %. All determinations were performed in triplicate.

### 4.14. Statistical Analysis

All statistical analyses were performed using Minitab 21 (Minitab LLC, State College, PA, USA). Differences among treatments were evaluated using one-way analysis of variance (ANOVA). When significant effects were detected (α = 0.05), Tukey’s multiple-comparison test was used to identify pairwise differences among means.

Multivariate analyses were conducted to examine relationships among compositional, technological, and digestibility variables. Principal Component Analysis (PCA) was used to reduce dimensionality and identify patterns of covariation among key attributes, including protein, phenolic, and dietary fiber fractions, firmness, and starch digestibility parameters. Hierarchical Clustering Analysis (HCA), based on Euclidean distance and Ward’s linkage method, was performed to classify the bread samples according to their multivariate profiles and to visualize groupings across storage days.

All analyses were performed using independent experimental replicates to ensure statistical robustness.

## 5. Conclusions

This study demonstrates that incorporating *C. aconitifolius* and *C. longirostrata* leaf flours into wheat bread formulations produces distinct and meaningful modifications in technological behavior, nutritional composition, and starch digestibility. Both ingredients increased protein, fiber, and phenolic content, contributing to nutritionally enhanced breads with a lower predicted glycemic index (pGI). *C. longirostrata* exerted the strongest effect on glycemic modulation, achieving the lowest pGI by Day 5, which was associated with its greater accumulation of resistant starch and slower starch hydrolysis. However, these nutritional advantages were accompanied by technological challenges, including reduced loaf expansion and a denser crumb structure, highlighting the need to balance functional benefits with processing performance.

In contrast, *C. aconitifolius* demonstrated superior technological compatibility, yielding breads with higher loaf volume, more cohesive crumb structure, and a moderate firming pattern during storage. Although its impact on pGI was less pronounced than that of *C. longirostrata*, the reductions observed, although modest, may still be biologically relevant, particularly for consumers sensitive to postprandial glycemic fluctuations. These findings underscore the differential roles of the two leaf flours and reinforce the importance of understanding ingredient-specific mechanisms rather than relying solely on extract-based evidence.

Overall, this work provides foundational insight into how whole leaf flours, with their complex fiber–phenolic matrices, interact with gluten and starch systems in bread. It also demonstrates that these botanical ingredients can contribute to the development of functional baked products with improved nutritional attributes and potentially lower glycemic impact, albeit with formulation-dependent technological trade-offs.

Future research should explore optimized substitution levels, evaluate blended formulations combining both flours, and assess processing adjustments, such as hydration management and mixing regimes, to improve dough handling and structural performance. Incorporating sensory evaluation, extended shelf-life characterization, and in vivo glycemic testing will further validate the applicability of these ingredients and support their integration into commercially viable functional bread products.

## Figures and Tables

**Figure 1 plants-15-00071-f001:**
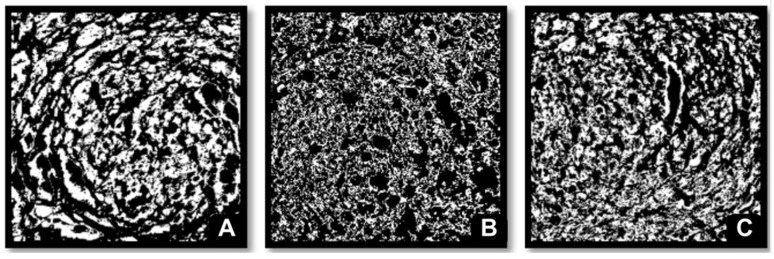
Crumb structure and alveolar distribution of control and plant-enriched breads. (**A**) Control, (**B**) *C. aconitifolius*, (**C**) *C. Longirostrata*.

**Figure 2 plants-15-00071-f002:**
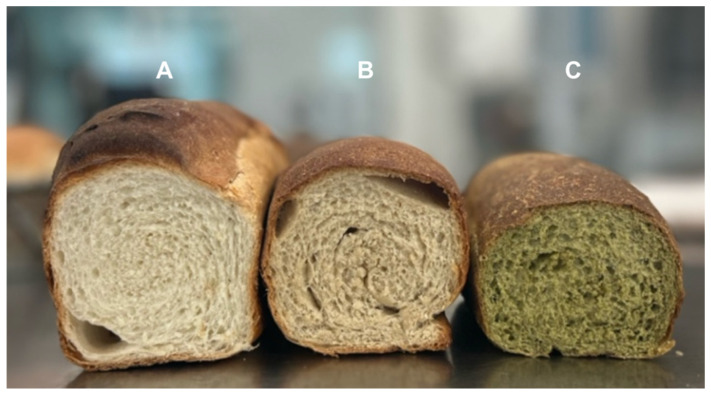
Cross-sectional appearance of control and leaf-flour-enriched breads: (**A**) control, (**B**) 5% *C. aconitifolius*, (**C**) 5% *C. longirostrata*.

**Figure 3 plants-15-00071-f003:**
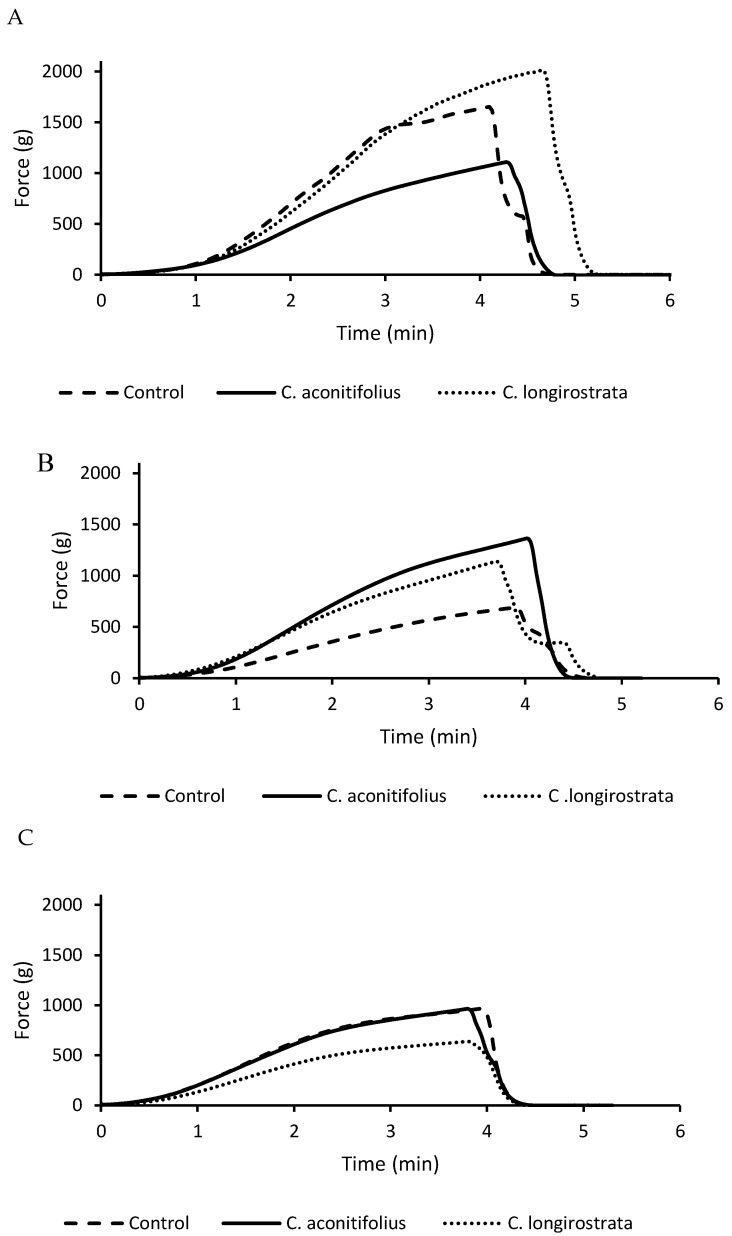
Firmness progression of bread crumb during storage (Day 0–5): (**A**) Day 0, (**B**) Day 3, (**C**) Day 5.

**Figure 4 plants-15-00071-f004:**
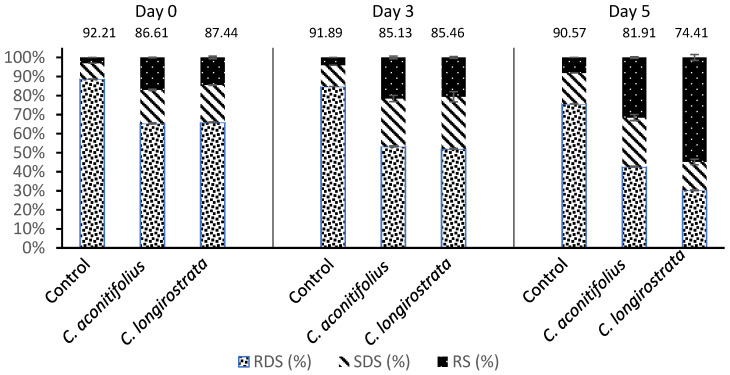
Starch digestibility profiles and predicted glycemic index (pGI) across storage. The number above each bar represents the predicted glycemic index value.

**Figure 5 plants-15-00071-f005:**
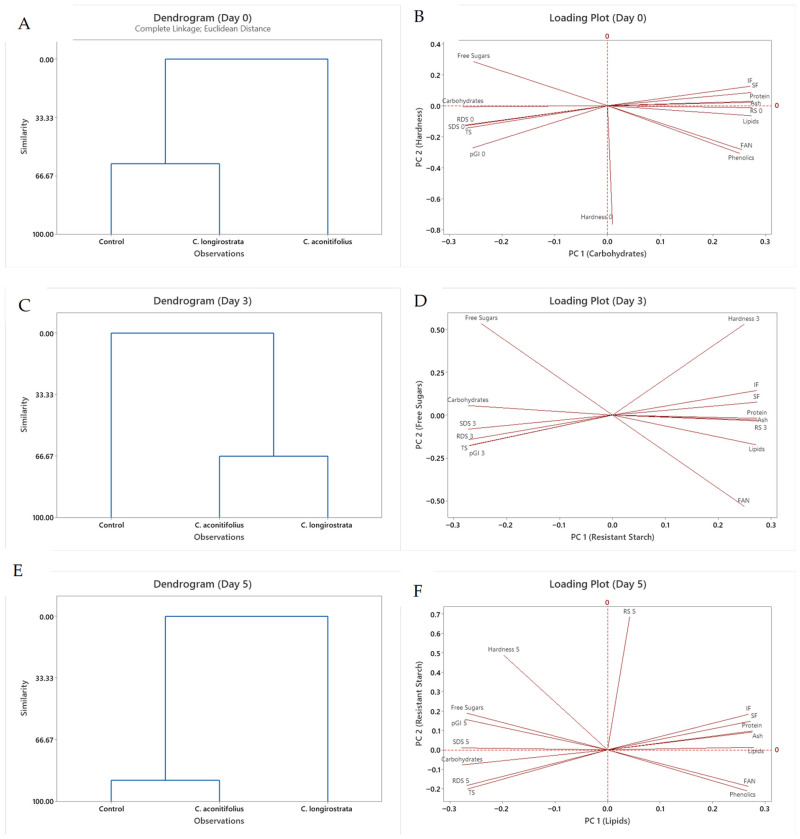
Multivariate analysis: dendrograms and PCA loading plots of bread samples. (**A**,**C**,**E**) Cluster analysis of samples. (**B**,**D**,**F**) Principal component analysis of selected variables.

**Table 1 plants-15-00071-t001:** Proximal composition of plant-enriched and control breads.

Sample	Moisture	Protein (%)	Lipids (%)	Ash (%)	TS (%)	SF (%)	IF (%)
Control	12.04 ± 0 a	10.13 ± 0.27 b	5.33 ± 0.11 c	5.44 ± 0.27 b	65.6 ± 0.30 a	1.26 ± 0.15 b	3.15 ± 0.13 b
*C. aconitifolius*	11.49 ± 0 b	12.24 ± 0.10 a	6.08 ± 0.49 b	6.24 ± 0.12 a	62.24 ± 0.20 b	2.22 ± 0.10 a	4.11 ± 0.09 a
*C. longirostrata*	11.44 ± 0 c	12.53 ± 0.10 a	6.32 ± 0.10 a	6.36 ± 0.17 a	62.3 ± 0.41 b	2.26 ± 0.08 a	4.09 ± 0.10 a

TS: total starch; SF: soluble fiber; IF: insoluble fiber. Values with different letters are significantly different (*p* < 0.05).

**Table 2 plants-15-00071-t002:** Phenolic content, free sugars, and free amino nitrogen of bread samples.

	Free Sugars (%)	FAN (mg/100 g^−1^)	Total Phenolics (mg GAE/100 g)
Control	11.5 ± 0.38 a	62.41 ± 1.96 b	0.01 ± 0.00 c
*C. acontifolius*	10.53 ± 0.36 b	68.31 ± 1.55 a	9.56 ± 1.21 b
*C. longirostrata*	9.54 ± 0.13 b	74.30 ± 2.18 a	19.82 ± 0.90 a

FAN: Free amino nitrogen. Values with different letters are significantly different (*p* < 0.05).

**Table 3 plants-15-00071-t003:** Physical parameters of breads: volume, height, apparent density, and crumb alveolation.

Sample	Height Before Baking (cm)	Height After Baking (cm)	Weight (g)	Volume (cm^3^)	Apparent Density (g/cm^3^)	Alveoli %
Control	6.5 ± 0.29 a	9.22 ± 0.81 a	146.65 ± 0.78 b	701.25 ± 42.69 a	0.20 ± 0.012 a	40.31 ± 1.33 a
*C. aconitifolius*	6.5 ± 0.08 a	7.92 ± 0.45 b	157.01 ± 4.81 a	597.5 ± 77.08 b	0.26 ± 0.031 ab	29.53 ± 0.64 b
*C. longirostrata*	5.55 ± 0.05 b	7.15 ± 0.19 c	153.25 ± 1.44 a	521.25 ± 11.08 b	0.29 ± 0.00 b	38.28 ± 2.02 a

Values with different letters are significantly different (*p* < 0.05).

**Table 4 plants-15-00071-t004:** Color attributes (CIE lab) and total color difference (ΔE) of breadcrumbs.

Sample	L*	a*	b*	ΔE
Control	78.42 ± 3.67 a	3.67 ± 0.11 b	16.43 ± 0.94 c	ND
*C. aconitifolius*	70.15 ± 1.97 b	7.55 ± 0.52 a	23.95 ± 0.90 b	70.01
*C. longirostrata*	51.88 ± 1.89 c	3.54 ± 0.12 b	28.12 ± 1.23 a	420.66

* Values with different letters are significantly different (*p* < 0.05). ND: Not determined.

**Table 5 plants-15-00071-t005:** Bread formulation with *C. aconitifolius* and *C. longirostrata* leaf flours.

Control		*C. longirostrata*		*C. aconitifolius*	
Ingredients	%	Ingredients	%	Ingredients	%
Wheat flour	100	Wheat flour	95	Wheat flour	95
Water	57	Water	61	Water	60
Sugar	6	Sugar	6	Sugar	6
Shortening	3	Shortening	3	Shortening	3
Compressed fresh yeast	6	Compressed fresh yeast	6	Compressed fresh yeast	6
Salt	2	Salt	2	Salt	2
Diastatic malt	0.3	Diastatic malt	0.3	Diastatic malt	0.3
		*C. longirostrata* flour	5	*C. aconitifolius* flour	5

Water was adjusted based on the type of flour used in the formulation.

## Data Availability

The raw data supporting the conclusions of this article will be made available by the authors on request.

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
