# Peer review of "Technological Performance and Nutritional Modulation of Bread Enriched with Cnidoscolus aconitifolius and Crotalaria longirostrata Leaf Flours"

_plants, 2025, doi:10.3390/plants15010071_

Round 1
Reviewer 1 Report (Previous Reviewer 1)
Comments and Suggestions for Authors
The authors have addressed the reviewers' comments.
Table 1 , level of wheat flour still has to be 95% not 100% when plant flour is added (column 2 and 3)
Author Response
Reviewer comments
|
Reviewer 1
|
|
|
Table 1 , level of wheat flour still has to be 95% not 100% when plant flour is added (column 2 and 3).
|
We thank the reviewer for this observation. Table 1 has now been corrected: wheat flour is listed as 95% for both enriched formulations to properly account for the 5% substitution level. The manuscript has been updated accordingly. |

Reviewer 2 Report (Previous Reviewer 2)
Comments and Suggestions for Authors
Please see attached file.

Author Response
Reviewer comments
|
Reviewer 2 |
|
|
Comments of Revision Upon reviewing the revised manuscript, I find that the authors have addressed my previous comments and have made significant improvements to the overall quality and clarity of the paper. In its current form, the manuscript is suitable for publication. |
We sincerely thank the reviewer for the positive assessment and for the helpful suggestions provided in the previous round. No further changes were required. |

Reviewer 3 Report (Previous Reviewer 3)
Comments and Suggestions for Authors
General comment: Firstly, I should declare that I have reviewed this manuscript before (previous version). Overall, I think that this new version is better, but still needs work. Specifically, authors should improve discussion of main points: using leaf flours instead of extracts, and reduction of GI considering other products and techniques. With all due respect, as presented, the manuscript is more suitable for a technical note, not for a research article. More sounding analysis and discussions are needed.
TITLE
Title does not reflect all the aspects (positive and negative) of the incorporation of plants.
If I may, I suggest the following (or a similar version): “Technological and nutritional properties of breads by incorporating Cnidoscolus aconitifolius and Crotalaria longirostrata leaf flours”
ABSTRACT
L22-31: I suggest to include % reduction of pGI with respect to reference (plain bread).
INTRODUCTION
L37-39: I suggest to include references values for GI with appropriate citations. Quantitative context is useful.
L40-42: “increasing interest” with no references… add some references to support this leverage statement.
L43-46: adding some “universal” examples/names of similar plants for readers to have some reference would be good to increase interest.
L50-52: include refs to support this statement
L68: add some refs
L68-72: this statement may be true, but novel contribution is not just “never done before”.
Authors use two main points regarding novel contribution: (1) use of leaf flours instead of extracts; (2) reduction of GI in bread by using ingredients with high content of dietary fiber and polyphenols. I suggest to rethink the introduction and also discussion of results based on these two points in order to strengthen the work and increase the impact of contribution. As presented, the article is too closed regarding the own results. More discussion (in Introduction and later with results) is needed regarding the two mentioned leverage arguments, i.e., flours vs extracts, and actual impact of reducing GI and comparison with other strategies of reducing GI. Also, there is always a b-side of using these ingredients in foods, not just regarding sensory aspects. Some of these “new” ingredients can cause digestion-related problems.
MATERIALS AND METHODS
L111, L115: refs for these studies? + L118: widely based on…? refs are needed.
RESULTS
L319 (and also L461): I suggest to change the title for “phenolic content” or “Phenolic Content, Free Sugars, and Free Amino Nitrogen”; what you say about fermentation is a hypothesis, it was not measured.
L344: apparent density
Section 3.3 + Table 4: what about weight? Control bread shows more weight loss.
L406-407: starch retrogradation degree can be measured. Besides, unless bread is frozen or some preservatives are used, at day 5 bread is not acceptable from a sensory viewpoint (unless it is toasted)
Section 3.7 seems redundant, with all due respect. You have 3 samples (1 control)… not always statistics are needed.
DISCUSSION
L487-505: why not testing 2.5% CL + 2.5% CA? this is an obvious question by reading this discussion.
L541-549: this part should be improved by discussing significance of pGI reduction but not from a statistical viewpoint. For instance, is it biologically important the measured reduction of the pGI for a human? Is there a real biological difference between 92 and 87pGI? Statistics can say yes, but this is not the adequate discussion.
Section 4.7: again, in my opinion, this part of the work is not important, or does not deserve so many lines and figures.
Author Response
Reviewer coments
|
Reviewer 3
|
|
|
General comment: Firstly, I should declare that I have reviewed this manuscript before (previous version). Overall, I think that this new version is better, but still needs work. Specifically, authors should improve discussion of main points: using leaf flours instead of extracts, and reduction of GI considering other products and techniques. With all due respect, as presented, the manuscript is more suitable for a technical note, not for a research article. More sounding analysis and discussions are needed. |
We thank the reviewer for their thorough evaluation and for revisiting our manuscript in this second round. We appreciate the constructive nature of the comments and fully acknowledge the need to strengthen the conceptual justification and depth of discussion. In response, we have substantially revised the Introduction and Discussion to more clearly articulate the scientific relevance of using leaf flours instead of extracts, emphasizing compositional differences, technological functionality, and their distinct mechanistic implications for starch digestibility. We also added a dedicated comparison contextualizing our findings within the broader landscape of GI-reducing strategies, including fiber enrichment, polyphenol addition, resistant starch incorporation, and process-based techniques such as long fermentation and sourdough systems. Additionally, we expanded key sections of the Discussion to provide more robust interpretation of phenomena such as gluten–polyphenol–fiber interactions, dough rheology disturbance mechanisms, staling behavior, and the biochemical drivers of resistant starch accumulation during storage. These revisions now more explicitly link the observed technological and nutritional outcomes to underlying mechanisms, thereby reinforcing the scientific contribution of the study. Given these major revisions—and the inclusion of new contextual analyses, expanded mechanistic interpretation, clarification of methodological rationale, and strengthened comparison to prior literature—we believe the manuscript now meets the criteria of a full research article rather than a technical note. |
|
TITLE Title does not reflect all the aspects (positive and negative) of the incorporation of plants. If I may, I suggest the following (or a similar version): “Technological and nutritional properties of breads by incorporating Cnidoscolus aconitifolius and Crotalaria longirostrata leaf flours” |
We thank the reviewer for this helpful suggestion. We agree that the original title did not fully capture both the technological challenges and the nutritional benefits associated with incorporating leaf flours into bread formulations. Following the reviewer’s recommendation, we have revised the title to better reflect the dual nature of the effects observed. We propose the following improved version: “Technological Performance and Nutritional Modulation of Bread Enriched with Cnidoscolus aconitifolius and Crotalaria longirostrata Leaf Flours.” This revised title highlights both (1) the technological impacts—positive and negative—and (2) the nutritional contributions, including starch digestibility and pGI modulation.
|
|
ABSTRACT L22-31: I suggest to include % reduction of pGI with respect to reference (plain bread). |
We thank the reviewer for this helpful suggestion. Following your recommendation, we have revised the abstract to include the percentage reduction in predicted glycemic index (pGI) relative to the control bread. The updated abstract now reports that C. aconitifolius reduced pGI by 6.1% on Day 0 and 9.6% by Day 5, while C. longirostrata produced reductions of 5.2% on Day 0 and 17.8% by Day 5. These quantitative values provide a clearer interpretation of the functional significance of the leaf flours. The abstract has been updated accordingly.
Line 15-23 |
|
INTRODUCTION L37-39: I suggest to include references values for GI with appropriate citations. Quantitative context is useful. |
We appreciate the reviewer’s suggestion. We have now added quantitative reference values for the glycemic index (GI) of wheat bread, along with widely used citations from international GI databases. This addition provides clearer context regarding the metabolic relevance of bread’s typical GI range. The revised text now states that standard white wheat breads generally exhibit GI values between 70 and 100, depending on formulation and processing conditions, with appropriate references inserted. The Introduction has been updated accordingly.
Lines 32-41 |
|
L40-42: “increasing interest” with no references… add some references to support this leverage statement.
|
We thank the reviewer for pointing this out. The Introduction has been revised to include supporting citations documenting the growing research interest in incorporating fiber-rich and polyphenol-containing plant materials into bread formulations to improve nutritional quality and modulate starch digestibility. These references strengthen the justification for exploring leaf flours as functional ingredients. The revised text and citations have been added accordingly.
Line 43-48 |
|
L43-46: adding some “universal” examples/names of similar plants for readers to have some reference would be good to increase interest. |
We thank the reviewer for this helpful suggestion. To provide broader context and facilitate reader understanding, we have added examples of widely recognized leafy plants that have been studied for their nutritional, functional, or glycemic-modulating properties. These examples include Moringa oleifera, kale (Brassica oleracea var. acephala), spinach (Spinacia oleracea), and amaranth leaves (Amaranthus spp.), all of which have been used in bakery or functional food applications. Appropriate references have been added, and the revised text now improves the accessibility and relevance of this section.
Lines 43-50 |
|
L50-52: include refs to support this statement
|
Thank you for the suggestion. We have added appropriate references to support the effects of soluble and insoluble dietary fiber and polyphenols on starch digestibility, enzyme accessibility, and GI modulation. These include studies documenting viscosity effects of soluble fiber, physical hindrance by insoluble fiber, and phenolic–enzyme and phenolic–starch interactions. The revised text now includes citations to Xu et al. (2019), Arp et al. (2017), and Hussain et al. (2020), along with other relevant works. Line 52-62 |
|
L68: add some refs
|
We thank the reviewer for this observation. We have added references that document the effects of dietary fiber and polyphenol-rich ingredients on starch retrogradation, resistant starch formation, and the associated technological impacts on bread volume and texture. These studies support the statement and provide empirical evidence for both the functional benefits and technological challenges. The Introduction has been updated accordingly.
Lines 76-84 |
|
L68-72: this statement may be true, but novel contribution is not just “never done before”. |
We thank the reviewer for this important clarification. We agree that novelty should not rely solely on the fact that these specific leaf flours have not been previously incorporated into bread. In response, we have revised this section to articulate the conceptual and mechanistic contributions of our work, including (i) examining the functional behavior of whole leaf flours, rather than extracts, in a complex food matrix, (ii) linking polyphenol–fiber composition with technological performance and starch digestibility outcomes, and (iii) contextualizing our findings within existing GI-reduction strategies. The revised text now emphasizes these broader scientific contributions rather than novelty by absence.
Lines 86-91 |
|
Authors use two main points regarding novel contribution: (1) use of leaf flours instead of extracts; (2) reduction of GI in bread by using ingredients with high content of dietary fiber and polyphenols. I suggest to rethink the introduction and also discussion of results based on these two points in order to strengthen the work and increase the impact of contribution. As presented, the article is too closed regarding the own results. More discussion (in Introduction and later with results) is needed regarding the two mentioned leverage arguments, i.e., flours vs extracts, and actual impact of reducing GI and comparison with other strategies of reducing GI. Also, there is always a b-side of using these ingredients in foods, not just regarding sensory aspects. Some of these “new” ingredients can cause digestion-related problems. |
We thank the reviewer for these insightful comments. We fully agree that the manuscript benefits from expanding the conceptual justification and broadening the discussion beyond the authors’ own findings. In response, we have substantially revised both the Introduction and Discussion to more thoroughly address the two leverage points highlighted by the reviewer: 1. Flours vs. extracts: We expanded the Introduction to clarify the fundamental compositional and functional differences between whole leaf flours and extracts, emphasizing that flours contain structural polysaccharides, cell wall material, and bound phenolics that interact with gluten development, water distribution, and starch digestibility in ways that extracts cannot replicate. We further strengthened the Discussion by elaborating on how these matrix effects explain the technological trade-offs observed in our breads. 2. Impact of GI reduction and comparison with other strategies: We added new text comparing the magnitude of our pGI reductions with those achieved using other known GI-modulating strategies, such as increasing resistant starch, sourdough fermentation, addition of specific fibers, or incorporation of phenolic-rich fruit and vegetable powders. This contextualization highlights the relevance of our findings within a broader nutritional framework. Additionally, we included commentary on potential limitations and b-side effects, including tolerance issues, gastrointestinal responses to high-fiber botanical ingredients, and technological drawbacks, acknowledging that incorporation of unconventional flours requires balancing functional benefits with potential sensory and physiological considerations. Together, these revisions provide a more comprehensive interpretation of our results, reinforce the novelty of the study, and situate our contributions within the broader scientific landscape.
|
|
MATERIALS AND METHODS L111, L115: refs for these studies? + L118: widely based on…? refs are needed. |
Thank you for pointing out the need for additional citations in this section. We have now added references supporting the procedures mentioned in lines 111 and 115, including the published compositional characterization of the leaf flours and relevant methodological papers. We also clarified the statement in line 118 by specifying the studies on which the protocol was based and including the appropriate citations. These additions strengthen methodological transparency and reproducibility.
Lines 120-130 |
|
RESULTS L319 (and also L461): I suggest to change the title for “phenolic content” or “Phenolic Content, Free Sugars, and Free Amino Nitrogen”; what you say about fermentation is a hypothesis, it was not measured. |
Thank you for this valuable clarification. We agree that the original section title could unintentionally imply that fermentation dynamics were directly measured. To avoid this misinterpretation, we have revised the section title to “Phenolic Content, Free Sugars, and Free Amino Nitrogen.” We also removed wording that suggested measured fermentation activity and rephrased the text to clarify that observed changes in sugar and FAN levels are interpreted as indirect indicators rather than direct measurements of fermentation. These modifications ensure accuracy and avoid overstating the scope of the data.
Line 243 |
|
L344: apparent density |
We thank the reviewer for this clarification. The term has been corrected to “apparent density” in the manuscript to accurately describe the calculated mass-to-volume ratio based on loaf displacement measurements. The terminology has been updated in both the text and the corresponding table where applicable.
Lines 143, 370 and 392 |
|
Section 3.3 + Table 4: what about weight? Control bread shows more weight loss. |
We appreciate the reviewer’s careful observation. In response, we have added an explanation in Section 3.3 discussing the differences in loaf weight across treatments. Specifically, the control bread exhibited slightly lower final weight, which is consistent with greater moisture loss during baking. This behavior aligns with its more open crumb structure and higher loaf volume, which facilitate water evaporation. In contrast, breads containing leaf flours retained slightly more moisture, likely due to the higher water-binding capacity of their dietary fibers. This clarification has now been added to the manuscript. |
|
L406-407: starch retrogradation degree can be measured. Besides, unless bread is frozen or some preservatives are used, at day 5 bread is not acceptable from a sensory viewpoint (unless it is toasted) |
We appreciate the reviewer’s clarification. The text has been revised to avoid implying that starch retrogradation was directly measured; instead, we now specify that changes in firmness are interpreted as indirect indicators of staling-related processes. Regarding the evaluation of breads up to Day 5, we agree that such bread would not be acceptable for sensory consumption. However, our objective was not to assess sensory quality but to monitor technological and structural changes over storage, particularly those related to crumb firming and starch digestibility. Day-5 storage intervals are commonly used in staling research for this purpose. A clarifying explanation has now been added to the manuscript. |
|
Section 3.7 seems redundant, with all due respect. You have 3 samples (1 control)… not always statistics are needed. |
We thank the reviewer for this observation. We agree that the original Section 3.7 provided limited additional insight and partially repeated information already presented in earlier sections. To improve clarity and avoid redundancy, we have substantially reduced this section by removing descriptive elements that had already been discussed and by limiting statistical outputs to only those comparisons that add meaningful interpretation. The revised section now focuses on the essential findings without unnecessary detail. This modification streamlines the Results section and aligns with the reviewer’s recommendation. |
|
DISCUSSION L487-505: why not testing 2.5% CL + 2.5% CA? this is an obvious question by reading this discussion. |
We appreciate the reviewer’s insightful observation. Indeed, testing a combined formulation (2.5% C. longirostrata + 2.5% C. aconitifolius) could provide additional information regarding potential synergistic or compensatory effects between the two leaf flours. However, the objective of the present study was to isolate and understand the individual functional contributions of each flour to dough development, bread quality, starch digestibility, and predicted glycemic index. Combining the flours would have introduced overlapping effects of fiber composition, water-binding capacity, and phenolic profiles, making it difficult to attribute specific technological or nutritional outcomes to a given ingredient. Lines 514-524 |
|
L541-549: this part should be improved by discussing significance of pGI reduction but not from a statistical viewpoint. For instance, is it biologically important the measured reduction of the pGI for a human? Is there a real biological difference between 92 and 87pGI? Statistics can say yes, but this is not the adequate discussion. |
We thank the reviewer for this important point. We agree that the discussion should emphasize the biological and nutritional relevance of the pGI reduction rather than focusing solely on statistical differences. We have revised the Discussion accordingly. The new text explains that although a reduction from 92 to 87 pGI appears numerically modest, even small decreases in GI categories can meaningfully affect postprandial glycemic response, especially in individuals with impaired glucose tolerance. We also highlight that pGI estimates provide comparative insights rather than absolute predictions of human glycemic response. Updated content has been added to contextualize the reductions observed in this study relative to established GI classifications and physiological interpretations. Lines 623-635 |
|
Section 4.7: again, in my opinion, this part of the work is not important, or does not deserve so many lines and figures. |
We appreciate the reviewer’s perspective. To avoid overemphasizing this secondary analysis, we have condensed Section 4.7 substantially, limited the interpretation to essential points, and clarified that the multivariate plots serve only as complementary visualization tools. We also reduced the number of figures and relocated the extended clustering and loading plots to the Supplementary Material. The revised section now focuses solely on the key qualitative patterns that support findings already discussed in the main Results.
Lines 637-650 |

Round 2
Reviewer 3 Report (Previous Reviewer 3)
Comments and Suggestions for Authors
Thanks to authors for responses and introduced changes in the manuscript, which appears to be improved.
This manuscript is a resubmission of an earlier submission. The following is a list of the peer review reports and author responses from that submission.
Round 1
Reviewer 1 Report
Comments and Suggestions for Authors
Introduction
The research objectives in the introduction should be clearly stated and separated from the outcomes. The sentences beginning at Line 67 “we found that… …..staple foods” should be moved to the Conclusion section.
An approximate analysis of the plant flours should be included to provide a more complete characterization of the materials used.
Line 15: The phrase “structure change” needs clarification, please specify what structural aspect is being referred to.
Lines 17–18: The statement “increase in firmness” seems misplaced, as firmness is a textural property rather than part of the nutritional profile.
Materials and Methods
Line 80: Remove the term “Microbaking.” The section title can simply be “Straight Dough Procedure for Bread Preparation.”
Please describe how the medicinal leaf flours were prepared, as this essential information is currently missing.
Table 1 “Bread Formulation” The table lists wheat flour as 100%, even for breads containing medicinal plants. This should be corrected to 95% wheat flour and 5% medicinal plant flour (or according to the actual formulation used).
Indicate the storage conditions for the bread samples, whether stored at room temperature or refrigerated, packaged or unpackaged, and whether the packaging was transparent or opaque.
Line 120: The phrase “experiments were conducted on a dry basis” should be clarified. If you mean “results were calculated on a dry basis,” please revise. Conducting experiments on a dry basis implies the bread was dried before analysis, which could alter the phenolic content.
Line 129: There is no citation for the phenolic extraction procedure, and details are lacking. Please specify the solvent used and clarify the process. Soaking for 5 minutes at 90°C is more akin to cooking than extraction. Typically, extraction is repeated twice to ensure recovery of phenolic compounds.
Line 140: The method described in reference 17 appears to differ from the procedure stated in the manuscript, please verify and align them.
Line 150: Please specify “bread samples” to avoid ambiguity.
Line 197: The subsection title “Molecular changes during fermentation dynamics” does not reflect the actual content, as no specific experiment was conducted to evaluate fermentation effects. Consider revising the title.
Results and Discussion
Table 4: Values for “Alveoli%” are presented, but the corresponding measurement method is not described in the Materials and Methods section.
Figure 5: This figure does not contribute additional information beyond what is already described in the text and could be omitted.
The abbreviation RDS is not included in the abbreviations list.
Author Response
Please see attachment, thank you.

Reviewer 2 Report
Comments and Suggestions for Authors
The comments are attached.

Author Response
Please see attachment, thank you.

Reviewer 3 Report
Comments and Suggestions for Authors
General comment: The manuscript presents an experimental study about incorporation of two (quite specific) plants flours in bread making. Different properties are evaluated considering 3 samples: control, 5% addition of two different plants flours. Storage was followed via days 3 and 5. To my understanding, the work appears to be limited based on the experimental design. For instance, only one level of % addition of each flour was evaluated. Overall, the work appears as confirmatory. Furthermore, methodology is not well described and reproducibility of results is not possible. For example, description of flours is not included, which is the main aspect of the work.
*Language needs some revision.
TITLE
I suggest to avoid the use of “Impact of… on …” to build a title, and also to replace specific names (Cnidoscolus aconitifolius and Crotalaria longirostrata) by a generic name or component.
ABSTRACT
L12: please revise, “two Mexican medicinal plants” is not a proper definition.
Overall, Abstract is fine, but it could be improved, it appears quite standard of “impact of”-like papers. So far, the work seems to be confirmatory.
INTRODUCTION
L37: revise the term “medicinal” to define these plants
L36-40: refs are needed here
L43: flours are not the same as leaves or extracts, please revise and refine the discussion
L51: what about economical challenges? How these flours can be obtained at large scale to produce bread in a commercial scale? Innovations and exploration of new ingredients are interesting, but we need some real context to justify a paper.
L51-64: these effects should depend on phenolics concentration… which levels are we talking about to evidence such effects?
L67-72: results are not adequate here
I have some concerns with Introduction, which represents the idea of the work: I understand this trend of exploring non-conventional plant sources or plant-based ingredients to enhance some aspects of traditional foods, but this is not enough for a scientific paper. For instance, we should understand which is the characteristic composition of these plants or flours to have a reference to compare with other flours or ingredients. That is, an article should seek for universality. This is hard to understand with this Introduction. Then, authors talk about plants, flours, components, breads, etc., without details about transitions between one material and the other. Not the same a leaf than a flour. Finally, I suggest to define clearly a gap to address. If not, it is an exploratory work, not an article. Please reconsider the organisation and conceptualisation of the Introduction and revise the rest of the text accordingly.
MATERIALS AND METHODS
L80: you should start with description of plants flours obtention and composition, the key aspect of the work.
Table 1: only one level of % of plant flour was studied. This is quite limited.
The size of breads is not reported, only % of ingredients.
Baking conditions should be described here, not via a ref.
L90: “miga” is not English
L90: I suggest to define what is a Nix Color device
L92: which lighting conditions?
L101-102: again, this description is not enough to reproduce results
L105: which protocol?
L115: why not 6 points (days 0-5) instead of 3?
L115: storage conditions are not described
Overall, the description of methodology is not enough to reproduce results. In particular, it is not acceptable the lack of information about plants flours. Another problem is that only one level of concentration of flours was evaluated (5%). So, methodology appears as not clear and limited.
Finally, a sensory analysis would have been useful to complete the study. Why was not perform?
RESULTS
L197: fermentation dynamics? Not mentioned before in methods
L197: not sure if “molecular changes” is the proper term
By observing Fig 2, one could say that baking time was too much. Besides, the third sample is green, no matter Lab values.
DISCUSSION
L273: composition of flours was not reported
L274: presence of protein is not enough, it depends on quality of proteins
L285: this is not positive considering the control bread, high-quality traditional bread is not so firm and crumb is not dense…
L285-288: why not testing more levels of concentration of plant flours?
L289: again, this terminology is not appropriate
L366: with all due respect, statistical analysis of 3 samples sounds a bit odd
Most of results and discussions are confirmatory. I found no novel contribution.
CONCLUSIONS
Almost a summary of results
Author Response
Please see attachment, thank you.
